# Microvascular Cortical Dynamics in Minimal Invasive Deep-Seated Brain Tumour Surgery

**DOI:** 10.3390/cancers17091392

**Published:** 2025-04-22

**Authors:** José Pedro Lavrador, Oliver Wroe-Wright, Francesco Marchi, Ali Elhag, Andrew O’Keeffe, Pablo De La Fuente, Christos Soumpasis, Andrea Cardia, Ana Mirallave-Pescador, Alba Díaz-Baamonde, Jose Sadio Mosquera, Domingos Coiteiro, Sharon Jewell, Anthony Strong, Richard Gullan, Keyoumars Ashkan, Francesco Vergani, Ahilan Kailaya Vasan, Ranjeev Bhangoo

**Affiliations:** 1Department of Neurosurgery, King’s College Hospital Foundation Trust, London SE5 9RS, UK; josepedro.lavrador@nhs.net (J.P.L.); oliver.wroewright1@nhs.net (O.W.-W.); ali.elhag@nhs.net (A.E.); a.okeeffe@nhs.net (A.O.); a.mirallave-pescador@nhs.net (A.M.-P.); a.diazbaamonde@nhs.net (A.D.-B.); jose.mosquera@nhs.net (J.S.M.); richardgullan@nhs.net (R.G.); k.ashkan@nhs.net (K.A.); francesco.vergani@nhs.net (F.V.); a.kailaya-vasan@nhs.net (A.K.V.); ranj.bhangoo@nhs.net (R.B.); 2Department of Neurosurgery, Neurocenter of Southern Switzerland, Ente Ospedaliero Cantonale, 6500 Lugano, Switzerland; andrea.cardia@eoc.ch; 3Department of Neurosurgery, Araba University Hospital, 01009 Vitoria, Spain; pablojoseba.delafuentevilla@osakidetza.eus; 4Department of Neurosurgery, The National Hospital for Neurology and Neurosurgery, London WC1N 3BG, UK; christos.soumpasis@nhs.net; 5Department of Neurophysiology, King’s College Hospital Foundation Trust, London SE5 9RS, UK; 6Unidade de Saude Local, Hospital Santa Maria, 1649-028 Lisboa, Portugal; dcoiteiro@hospitaldaluz.pt; 7Department of Basic and Clinical Neuroscience, Institute of Psychiatry, Psychology and Neuroscience, King’s College University, London WC2R 2LS, UK; sharon.1.jewell@kcl.ac.uk (S.J.); anthony.strong@nhs.net (A.S.)

**Keywords:** transsulcal approach, minimal invasive parafascicular surgery, neuro-oncology, tubular retractor, microvascularization, deep-seated lesions

## Abstract

The minimally invasive parafascicular approach for the removal of deep-seated brain tumours represents a challenge in neurosurgical oncology. The use of tubular retractors with a transsulcal cannulation is the safest technique to reach those lesions minimizing the risk of brain injury and postoperative neurological deficit. As far as we know, no reports in the literature are present concerning the impact on cortical vascularization, its dynamics and the clinical consequences in terms of the patient’s outcome. Therefore, we assess the impact of tubular retractors in the microvascular dynamics on the surrounding brain using a quantitative indocyanine green (ICG)-based module in the operative microscopes and we correlate these measurements with neurological outcome and postoperative MRI restriction to diffusion areas. Our study demonstrates an increase in speed and cerebral blood flow index related with worse neurological presentation and postoperative neurological outcome in both pre-cannulation and pre–post-cannulation analysis, making it a strong predictor of worse outcome.

## 1. Introduction

Onco-functional balance is the current gold standard in neuro-oncology [1,2]. Maximizing surgical resection whilst preserving neurological function is a difficult balance in eloquent lesions [3,4,5]. This becomes a significant challenge when approaching deep-seated lesions due to the potential injury of non-involved cortical and subcortical brain tissue [6,7]. In this context, the transsulcal tubular retractor-assisted minimally invasive parafascicular approach (ts-trMIPS) has gained significant popularity as it is a tissue-sparing technique preserving cortical and subcortical structures from the brain surface to the lesion [8,9,10,11,12]. As a transsulcal approach, it potentiates the preservation of the main projection and association fibers that originate from the crest of the gyrus. This benefit is combined with stable retraction and pressure over the surrounding tissue provided by the tubular retractor. Previous publications have shown a decrease in the chances of perisurgical corridor contusion and laceration when compared with blades or brain spatulas [6,7]. However, no study has assessed the impact of this technique on the surrounding cortical vascularization. This is critical, as sparing brain tissue finds its parallel in the functional state left for that tissue to perform.

Indocyanine green videoangiography (ICG-VA) is a well-established intraoperative technique to assess vascular anatomy and undertake flow assessment. It was first approved by the FDA in 1959 for intraoperative chronic liver disease, and after significant expansion in the ophthalmology field at the end of last century, it has been used extensively in cranial vascular surgery since 2003 [13,14]. Recent analytic algorithms further expanded its field from an anatomical tool towards a functional assessment adjunct that has revolutionized the understanding of the vascular network [15,16,17,18,19]. As a consequence, the main focus of ICG–videoangiography application in neurosurgery has been the vascular surgery pathology, such as aneurysms [20,21,22], arteriovenous malformations [23,24], dural artero-venous fistulas [25,26], and revascularization surgery [27,28]. Fluorescence-based metrics (maximal fluorescence and luminosity), time-based metrics (delay, time to peak and rise time) and mixed metrics (cerebral blood flow index) provide relevant information about potential vascular compromise and regional blood flow [16].

Less is known about the potential implications of ICG-VA quantification in the neuro-oncology field. Its use has prioritized the anatomical definition/preservation of the vascular structure [15,29,30,31,32] or assessment of the vascular impact of tumour resection in draining veins [33,34]. Quantitative analysis has also been reported for intraoperative decision of a revascularization procedure after inadvertent occlusion of a middle cerebral artery branch with intraoperative drop in motor-evoked potentials [35]. However, no case series have systematically assessed the correlation between quantitative ICG-derived metrics and neurological outcome, particularly in the field of tubular retractor surgery.

Therefore, this study aims to assess the impact of ts-trMIPS in the peri-surgical corridor microvascularization and its correlation with lesion- and surgical-specific factors and postoperative clinical and radiological outcomes. We will use an ICG-VA quantification approach to understand the implications of this surgical technique on the surrounding blood flow.

## 2. Materials and Methods

This is a single-centre retrospective cohort study of adult patients admitted for surgery of a deep-seated intrinsic brain tumour between 2019 and 2023. The inclusion criteria were the ts-trMIPS surgical technique (BrainPath System^©^ Nico Corporation, Indianapolis, IN, USA) and pre- and post tumour resection ICG-VA with FLOW 800 analysis using a ZEISS KINEVO 900^©^ microscope (Kinevo 900 Zeiss^©^, Pentero, Carl Zeiss AG, Oberkochen, Germany). The exclusion criteria were age < 18 years old, conversion to non-tubular retractor assisted surgical technique, transgyral approach and incomplete ICG-VA FLOW 800 data analysis. All patients consented to this surgical procedure and the use of ICG-VA for vascular assessment in a standard neurosurgical adjunct.

The surgical procedure for the patients included in the study followed a standard approach. Preoperative volumetric structural brain imaging using T1-weighted contrast enhanced sequencing was used for intraoperative navigation. A diffusion-weighted image (64 directions) was used for preoperative tractography for subcortical corridor planning. Navigated transcranial magnetic stimulation was used to assess the cortical motor and language function and determine a safe entry point. Three-dimensional modelling of the tumour, cortical functional mapping, subcortical structural tractography and the ts-trMIPS approach were performed using Cranial Stealth Station Medtronic^©^ Software S8. Intraoperatively, all patients underwent intraoperative mapping and neuromonitoring guidance for motor, language or visual functional according to the location of the tumour. After the initial opening stages, the sulcus of interest was opened, and the first ICG-VA FLOW 800 assessment was performed. Thereafter, the tubular retractor was introduced—brain cannulation—towards the lesion according to a surface docking technique using optical neuronavigation (Medtronic^©^). A NICO BrainPath^©^ tubular retractor system was used in all included patients with a standard diameter of 13.5 mm. The surgical resection was performed using bipolar forceps, an ultrasonic aspirator and a NICO Myriad^©^ microdebrider (NICO Myriad System^©^ Nico Corporation Indianapolis, IN, USA) that was electrified as per a previous description [12]. After decannulation, a final ICG-VA FLOW 800 assessment was performed. The microscope was kept in a similar position in both assessments. The surgery was then concluded with the regular steps for closing (Figure 1).

We performed a *region-of-interest-based* (ROI) and *combined-ROI per patient-based* analysis. The same ROIs were selected in the peri-surgical corridor arteries for the FLOW 800 analysis before cannulation and after decannulation. To guarantee that the same vascular areas were analyzed, we maintained the same ROIs used in the pre-cannulation assessment for the post-decannulation assessment and performed visual inspection of the intraoperative images to ascertain they were in the same position to account for brain shift and deformation phenomena that occur in tumour surgery after resection. The location of the selected ROIs was determined by the chosen surgical corridor (sulcus of interest and perisulci arteries available to monitor). The number of ROIs selected was also different and determined by the size of the craniotomy which showed variability according to the other surgical adjuncts planned to be used during surgery (intraoperative ultrasound and intraoperative neuromonitoring and mapping tools, mainly).

The following ICG-VA-derived metrics were assessed and the percentage of the difference between both assessments per ROI was calculated:(a)*Delay*: the time interval from 0 to 50% of maximum fluorescence intensity.(b)*Speed*: velocity increase in fluorescence intensity during an observation period.(c)*Time to Peak* (TtP): from the appearance of fluorescence to maximum fluorescence intensity.(d)*Rise Time* (RT): time during which fluorescence intensity rises from 10 to 90% of its peak.(e)*Maximal Fluorescence*: maximal intensity measured in arbitrary intensity units.(f)*Cerebral Blood Flow Index* (CBFI): ratio of maximum fluorescence intensity to rise time.

For the *combined-ROI per patient-based* analysis, a mean of the calculations for each patient’s ROIs was performed. This combined-ROI per patient-based analysis aimed to provide robustness to the region-of-interest-based approach, allowing us to minimize the effect of outliers in the later approach.

Demographic (age and sex, clinical information), neurological examination before and after surgery, lesion-specific characteristics (initial volume, residual volume, extent of resection, EoR), WHO grading, lesion-depth and peri-corridor ischemia, and surgical-related characteristics (time of cannulation and blood pressure) were assessed and compared with the ROI-specific variables. The neurological outcome was classified as “deterioration”, “stable” or “improvement”. This classification took into consideration the motor grading according to the medical research council scale for muscle strength [36], language assessment according to the Sheffield Screening Test [37], and visual field assessment.

We used the Stata 13.1 software^©^ package (StataCorp LLC, Lakeway Drive College Station, TX, USA) to perform the statistical analysis. A *p* value < 0.05 was considered significant. The FLOW 800-derived variables and the percentage of their differences were treated as continuous variables and linear regression models were applied. A similar approach was performed for EoR, tumour and ischemia volumes, blood pressure and cannulation period. On the other hand, neurological deficit was considered a dichotomic variable and logistic regression models were applied.

We performed different adjustments for confounding for pre-cannulation, post-cannulation and pre–post-cannulation analysis. The mean arterial pressure was the main confounder considered in all analysis. For pre-cannulation, we considered the tumour-related mass effect on the cortex, and we adjusted for tumour volume and depth of the lesion. For the post-cannulation analysis, we considered the ischemia volume in the postoperative scan and the residual tumour volume as the main factors we had to consider for neurological outcome and mass effect and for the pre–post-cannulation analysis, we considered all the previous factors. Finally, we performed a subgroup analysis according to the median time of cannulation to assess the impact produced by the period of the time the cortical–subcortical structures were under the physical stress of the tubular retractor on the neurological and radiological outcomes.

## 3. Results

Forty-two patients were operated on using the ts-trMIPS technique during the inclusion period; seven patients were excluded due to absent (five patients) or incomplete (two patients) ICG-VA data. One hundred and forty-four regions-of-interest (ROIs) from the 35 included patients (1.33 male/female ratio; mean age 54.94 ± 15.32 years) were evaluated (average 4.11 ± 1.57 per patient, range 1–7). The majority of the lesions were located on the dominant hemisphere (60.00%) and on the frontal (34.29%) and temporal (22.86%) lobes. The main tumour histologies were WHO grade 4 glioblastoma (51.43%) and metastasis (25.71%). Of the total, 37.14% of patients had a preoperative neurological deficit and 71.43% of patients remained stable or improved after surgery. At follow-up, 25.71% of patients had permanent focal neurological deficit, a reduction of 30.78% when compared with the preoperative assessment (Table 1).

The mean path-related and overall ischemia were 5.16 ± 6.76 cc and 12.46 ± 20.72 cc, respectively (Table 1). No statistical significance was found between the amount of ischemia in the postoperative imaging and the neurological outcome (path-related ischemia *p* = 0.9688, overall ischemia *p* = 0.3794). In both the ROI-based or combined-ROI per patient analysis, the differences in the flow metrics assessed were not related with both path-related and overall ischemia (*p* > 0.05). An increase in both speed (*p* = 0.021) and CBFI (*p* = 0.039) were related with a larger path-related ischemic volume in the univariate analysis in the combined-ROI per patient analysis. However, this significance was not sustained after adjustment for MAP at time of decannulation, residual tumour volume, distance tumour-to-surface and time of cannulation (speed *p* = 0.083 and CBFI *p* = 0.104). A subgroup analysis according to the time of brain cannulation (cut-off: median of distribution) showed no statistically significant impact of time of brain cannulation in both new focal neurological deficit after surgery (*p* = 0.915) and restriction to diffusion around the surgical corridor (*p* = 0.176). With regard to the ICG-VA-derived metrics, the single ROI analysis showed a decrease in speed (*p* = 0.044) and TtP (*p* = 0.039) post-decannulation and an increase in the delay (*p* = 0.038), comparing pre-cannulation and post-decannulation in the group of patients with longer cannulation times. However, the combined-ROI per patient analysis did not confirm these findings (*p* > 0.05) (Appendix A).

### 3.1. Neurological Outcome

#### 3.1.1. Region-of-Interest-Based (ROI)

A total of 144 ROIs were selected (mean 4.11 ± 1.57 per patient). Initial flow assessment prior to brain cannulation showed increased speed (*p* = 0.0044) and CBFI (*p* = 0.0085) and decreased delay (*p* = 0.0147), TtP (*p* = 0.0005) and RT (*p* < 0.0001) in patients with preoperative neurological deficit (Table 2 and Figure 2).

After adjusting these findings for confounding of mean arterial pressure (MAP), distance from tumour-to-surface and tumour volume, speed (*p* = 0.001), TtP (*p* < 0.0001) and RT (*p* < 0.0001) remained significant. (Appendix A).

Final flow assessment after decannulation showed an increase in delay, decrease in speed, increase in RT, and a decrease in CBF in patients with neurological improvement after surgery (Table 2 and Figure 2). After adjustment for MAP at decannulation, residual tumour volume and ischemia, TtP (*p* = 0.022), RT (*p* = 0.007) and CBFI (*p* = 0.006) remained significant (Appendix A).

An increase in speed (deterioration = 43.12 ± 80.60% versus stable = −14.51 ± 57.80% versus improvement = −36.93 ± 31.33%, *p* < 0.0001) and CBFI (deterioration = 50.40 ± 88.17% versus stable = −2.70 ± 67.54% versus improvement = −38.98 ± 26.17%, *p* = 0.0005) and decrease in TtP (deterioration = −20.16 ± 27.33% versus stable = 34.73 ± 143.83% versus improvement = 14.20 ± 32.31%, *p* = 0.0005) were associated with a worse neurological outcome (Table 2 and Figure 2). Pairwise multiple comparison tests showed that the significant increase in speed and TtP and decrease in CBFI occurred when compared patients with neurological deterioration versus those with both stable (speed—*p* ≤ 0.0001 and CBFI—*p* = 0.006) or improved (speed—*p* ≤ 0.0001 and CBFI—*p* = 0.001) neurology. The main difference in TtP (*p* = 0.035) and RT (*p* = 0.049) occurred in the patients that showed neurological deterioration when compared with patients with stable neurology. After adjustment from pre- and post-cannulation MAPs, initial and residual tumour volume, distance tumour-to-surface and volume of ischemia in postoperative imaging, both an increase in speed (*p* < 0.0001) and increase in CBFI (*p* = 0.001) remained significant for neurological outcome (Appendix A).

A Receiver Operating Curve (ROC) analysis was performed considering a predictive model for new FND after surgery based on the flow changes related with the neurological outcome—increase in speed and CBFI and a decrease in the TtP. The combined predictive model showed an area under the curve (AUC) of 0.7529, which was favourable compared to each of the variables independently (AUC speed = 0.7339 versus AUC CBFI = 0.6893 versus AUC TtP = 0.6457), showing 69.57% sensitivity, 82.93% specificity, 53.33% positive predictive value (PPV) and 90.67% negative predictive value (NPV) when all three characteristics were present (Figure 2).

#### 3.1.2. Combined-ROI per Patient Analysis

Apart from an improved neurological outcome with an increase in the post-decannulation delay in the univariate analysis (*p* = 0.0173), no other single-point analysis showed a correlation between the flow metrics and the neurological outcome in both the uni- and multivariate analysis (*p* > 0.05) (Table 3 and Figure 3).

An increase in speed (deterioration = 45.33 ± 94.28% versus stable = 21.31 ± 46.96% versus improvement = −36.09 ± 30.28%, *p* = 0.0208) and CBFI (deterioration = 55.17 ± 77.45% versus stable = −14.13 ± 47.43% *versus* improvement = 38.39 ± 22.19%, *p* = 0.0212) after decannulation is significantly associated with worse neurological outcomes. Pairwise multiple comparison tests showed that a significant increase in speed (*p* = 0.029) occurred in the patients that showed neurological deterioration when compared with patients with stable neurology, whilst an increase in CBFI occurred when compared with both patients with stable (*p* = 0.046) and improved (*p* = 0.041) neurology. After adjusting for both pre- and post-cannulation MAPs, initial and residual tumour volume, distance tumour-to-surface and volume of ischemia in postoperative imaging, both an increase in speed (*p* = 0.041) and an increase in CBFI (*p* = 0.019) remained significant for neurological outcome (Table 3 and Figure 3).

An ROC analysis was performed considering a predictive model for new FND after surgery based on the flow changes related with the neurological outcome—increase in speed and CBFI—that showed a better area under the curve (AUC) = 0.8380 when compared with either speed (AUC = 0.7800) or CBFI (AUC = 0.7778) increase in isolation, with 83.33% sensitivity, 88.89% specificity, 71.43% PPV and 94.12% NPV when both characteristics were present (Figure 3).

## 4. Discussion

Flow assessment-derived metrics were related with neurological presentation and outcome in tubular retractor-assisted transsulcal minimally invasive parafascicular surgery for tumour resection. In the ROI-based analysis, an increased speed and CBFI and decreased delay, TtP and RT prior to brain cannulation were related with worse neurological presentation. After decannulation, an increase in delay, decrease in speed, increase in RT, and decrease in CBF were observed in patients with neurological improvement. These findings remained true for post-decannulation delay in the combined-ROI per patient analysis. The pre-post cannulation analysis showed an increase in speed and CBFI and decrease in TtP (ROI-based only) related with a deterioration in the patient’s neurology after surgery.

We decided to perform two different approaches for data analysis: ROI-based and combined-ROI per patient analysis. This decision was taken to maximize the information taken from each ROI—ROI-based analysis—but to decrease the impact of microscope position and illumination of the operating field (not necessarily the same conditions at both pre- and post-cannulation assessments)—for the combined-ROI per patient analysis. Also, ROI-based analysis can provide immediate information to the surgical team to be incorporated in the decision-making process, whilst a combined-ROI per patient requires more data extraction and analysis. Therefore, both are complimentary in terms of immediate use (ROI-based) or more accurate and less prone to focal/local bias (combined-ROI per patient).

An increase in speed and CBFI were consistent metrics related with worse neurological presentation and outcome after surgery (pre-cannulation assessment, post-cannulation, difference between pre- and post-cannulation and in both ROI-based and combined-ROI-based analysis). These findings were observed in animal studies in penumbra areas where the capillaries that remained perfused after an ischemic insult developed vasoconstriction and increased flow as a consequence of compensatory arterial vasodilation [38,39]. Even though they were less consistent across the analysis model used, the associations observed in the peri-ischemic tissue described between TtP, delay and RT and worse neurological outcome support this model.

We found no correlation with the ischemic volume around the cannulation path or total volume of ischemia. Despite lack of complete understanding about the mechanisms underlying this phenomenon, it is well-established that there is a lack of substantial functional collateral flow within the microvasculature connecting the pial arterioles and the subsurface microvessels [40,41]. Even though the subsurface flow is not significantly affected by a single vessel occlusion within the network, this will cause cortical subcortical ischemia, which supports our findings of dissociation between blood flow assessment and ischemia in the postoperative MRI scan.

Impaired neurovascular coupling in the peri-ischemic area may play a crucial role in the poor outcome even in the absence of ischemia [38,42,43]. Driving changes in cortical microvascular flow dynamics could also be occult spreading depolarizations (SDs)—well known to incite prolonged changes in the local microcirculation [44,45]. SDs are known to correlate with the appearance of delayed neurological deficits after subarachnoid hemorrhage [46] and they have recently been documented in high-grade glioma [47]. Frequent eruption of SDs in the peritumoral region has been reported experimentally, and they have thus accordingly been proposed as a potential mechanism underlying transient neurological deficits in this group [48]. Eloquent cortices may further be especially susceptible to their occurrence [49]. A hypothesis is that the increased flow in the cortical vessels acts as a surrogate marker for the flow dynamics in the capillary bed, reflecting more erythrocytes passing through the capillaries in a same amount of time, being responsible for a decrease in the capacity for oxygen extraction and functional shunting [43]. The main difference between these findings derived from stroke literature and the ones reported in this study from neurosurgery brain tumour resections via tubular retractors is the temporary insult provided by the tubular retractor and/or tumour resection that is prone to reversibility. The reversibility of neurovascular decoupling was reported in other pathologies [50] and render this a therapeutic opportunity in this population.

The ROC analyses provide significant prognostic information for neurological deterioration after surgery. Both ROI-based and combined-ROI-based analyses have high specificity and NPV, which translates into an ability to predict patients with no postoperative deterioration. The ROC for the combined-ROI-based analysis also has a high sensitivity and PPV, which allows for the prediction of neurological deterioration. Therefore, strategies to improve local perfusion, such as blood pressure manipulation, vasodilation target therapy or increased oxygenation, can be initiated to reverse the cerebral blood flow changes and potentially improve outcomes.

No tubular retractor-specific metrics were related with the microvascular flow metrics assessed (entry point, length of retractor). Even though the single-ROI analysis showed some trends between the time of cannulation and ICG-VA-derived metrics, none of these were confirmed at the patient level and, therefore, they are potentially not relevant in driving the observed micro- and macrovascular changes. This supports a lack of deleterious effect of the tubular retractor in the surrounding tissue, particularly the lack of significance of time-sensitive variables. This has been suggested for both subcortical [51] and cortical structures if a transsulcal approach is performed [52]. From a technical perspective, this study comes alongside others from the neurovascular field that support that ICG-VA FLOW 800 assessments can identify and provide quantitative metrics for both micro- and macrovascular changes after successful intraoperative treatment [53]. We hypothesize that these results can be reproduced in other types of neuro-oncology brain surgery where tubular retractors are not involved, pending future research on this topic. Particularly, it is important to understand how these metrics behave in superficial non-tubular retractor-assisted surgery. This will enhance our comprehension about the main drivers for these reported changes: tubular retractor surgical adjunct versus tumour location.

This population has highly eloquent tumours given the number of patients with preoperative neurological deficits. This further validates the associations presented between the microvascular flow dynamics and the functional outcome as the cortical and subcortical tissue in the vicinity of the surgical corridor used was neurologically significant as well. Therefore, the cortical (flow-based) and subcortical (ischemia-proven) observed changes were less likely to be silent.

## 5. Limitations and Strengths

This is a retrospective study with inherent limitations for this study design, particularly incomplete data that lead to patient exclusions from the analysis and variability in the number of ROIs selected per patient (this is the reason why we present the average and range of ROIs per patient). Also, from a technical perspective, data with regard to the distance between the microscope and the brain were not recorded and therefore this was not accounted for in the analysis. The flow assessment was performed using only one quantitative software (FLOW 800) and it was not reproduced or cross-validated with other software available on the market (not available in our department). The lack of standardization is a well-recognized problem in quantification-based approaches [17] and further work to mitigate this limitation is required to promote the widespread application of these results. Only cortical microvascularization was analyzed and no data reported to the subcortical compartment/tumour bed are provided in this study. The reasons behind this decision were as follows: (a) the lack of previous data reporting/validating subcortical perfusion assessments using FLOW 800 ICG-VA analysis; (b) the technical challenge of providing depth focus to acquire ROIs in cortical and subcortical areas in the same ICG analysis (which would imply multiple administrations and the potential effect of cross ICG contamination in between them); and (c) the artificial effect of the presence of the tubular retractor when acquiring subcortical data, which would challenge the significance of the results obtained. Despite our best efforts to adjust our analysis for tumour-specific effects, such as initial and residual tumour volumes and tumour-to-surface distance, the tumour itself is responsible for neurovascular uncoupling due to its effect on the extracellular matrix, vascular and perivascular cell environment [54]. Therefore, we cannot exclude that some of the observed changes pre- and post-cannulation are related to the impact of the tumour and/or its resection in the perisurgical tissue and not necessarily the insult produced by the tubular retractor. Further studies should address the particular impact of different histologies and tumour characteristics in these flow metrics, controlling for the other confounders. Also, this study was not powered enough to perform a subgroup analysis according to the specific type of neurological deficit and its postoperative evolution (motor, speech and visual). Large cohort studies would provide more information to build on the one provided in this manuscript to provide further granularity to the neurological outcome data. Finally, no immediate postoperative perfusion imaging was acquired as these sequences are not included in the protocol for brain tumour evaluation at this time. Future studies should consider these sequences and their correlation with the studied ICG-VA-derived metric and ascertain if they provide a better picture of the intraoperative findings when compared with diffusion imaging.

This is the first study in the literature to our knowledge that analyses the impact of microvascular cortical dynamics in tubular retractor-assisted transsulcal minimally invasive parafascicular surgery on neurological outcomes. ROI-based findings (validated by the combined-ROI per patient approach) can provide an easy and assessable tool for the surgical team to integrate in multimodal risk stratification methods of neurological outcome after surgery (as, for example, intraoperative neuromonitoring and preoperative mapping data). It is important to validate these observations with other tubular retractors available to establish the further reproducibility of these findings. Also, it is important to further understand the correlation of these intraoperative adjunct with other tools available for micro- and macrovasculatization assessment, such as the ecodoppler, as previous studies have failed to provide a reliable correlation [53]. As tubular retractor-assisted surgery gains traction in the surgical treatment of deep-seated lesions, a holistic understanding of its impact on cortical vascular dynamics is crucial to pave the way for future work focusing on treatment of these prognostic indicators of postoperative neurological function. These data can also improve patient selection for future studies using intra and/or postoperative treatments targeting the impact of cerebral blood flow and neurovascular coupling—euvolemia or flow augmentation via manipulation of postoperative targets for blood pressure management—on neurological outcomes.

## 6. Conclusions

Microvascular cortical dynamics are related with both neurological deficit at presentation and postoperative functional outcome in the transsulcal minimally invasive parafascicular approach. No tubular retractor-specific metric was related with flow changes in the surrounding brain tissue or neurological outcome. A single time-point analysis proved to be significantly related to neurology at presentation only in the ROI-based analysis but failed to sustain significance in the combined-ROI per patient analysis. Overall, an increase in speed and CBFI were related with worse neurological presentation and postoperative neurological outcome in both the pre-cannulation and pre–post-cannulation analysis in both the *ROI-based* and *combined-ROI per patient* analysis in the unadjusted and adjusted analysis, making them strong predictors of worse outcome.

## Figures and Tables

**Figure 1 cancers-17-01392-f001:**
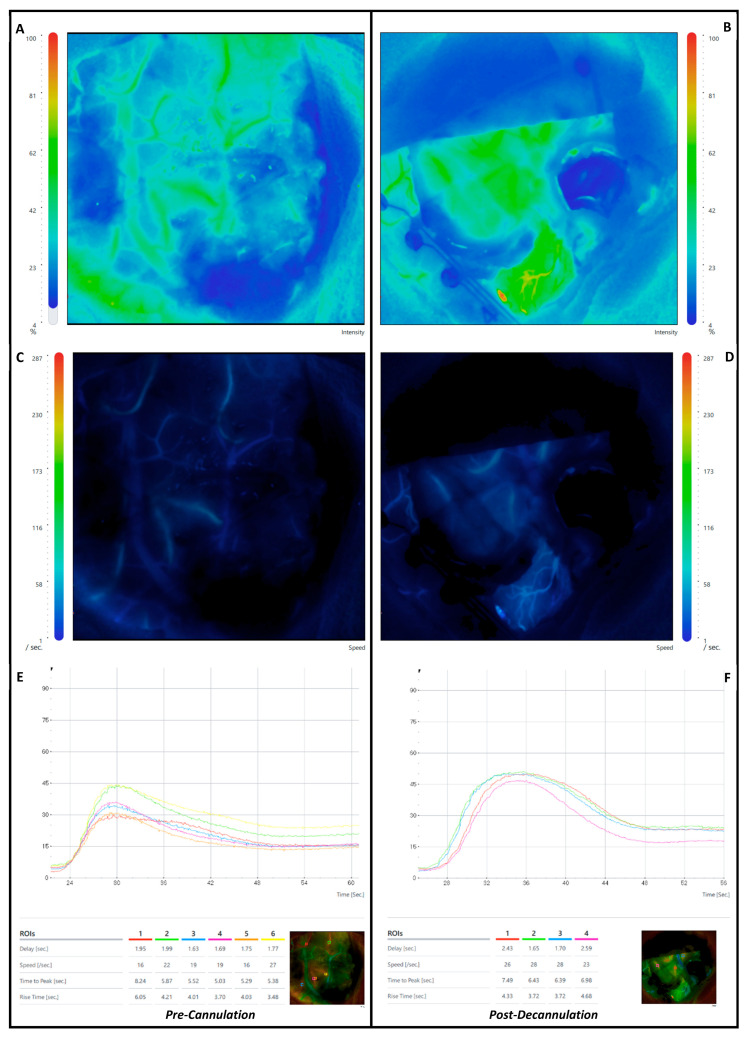
Indocyanine green quantitative assessment with Zeiss Flow 800^©^ module using Zeiss Kinevo 900^©^ Microscope—(**A**–**D**) show the pre-brain cannulation and post-brain decannulation intensity (**A**,**B**) and speed (**C**,**D**)—pre-cannulation on the left and post-cannulation on the right. (**E**,**F**) show the final diagram that assesses the delay, speed, time to peak and rise time in the selected regions-of-interest before (**E**) and after (**F**) brain cannulation.

**Figure 2 cancers-17-01392-f002:**
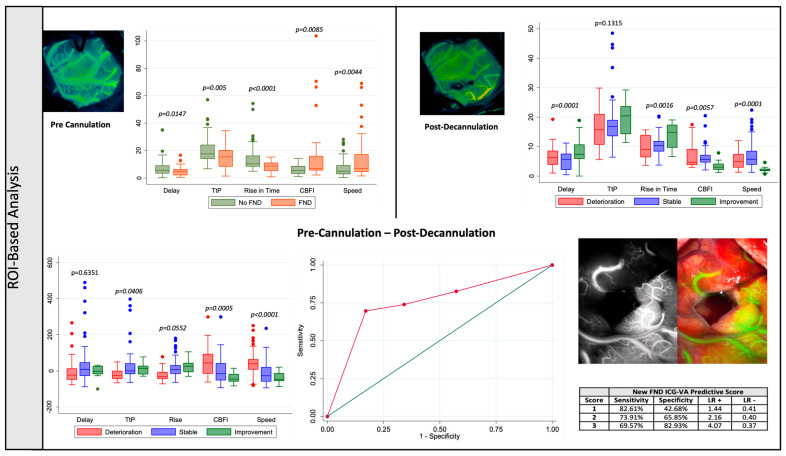
Region-of-Interest Based Analysis—(**Top left**): Pre brain cannulation quantitative flow metrics and presence of preoperative focal neurological deficit. (**Top right**): Post brain decannulation quantitative flow metrics and the postoperative overall neurology. (**Bottom left**): Pre brain cannulation—post brain decannulation ICG flow metric percentage changes and overall neurology. (**Bottom right**): Receiver Operating Characteristic Curve Analysis using the flow metrics that showed statistical significance during the uni and multivariate analysis.

**Figure 3 cancers-17-01392-f003:**
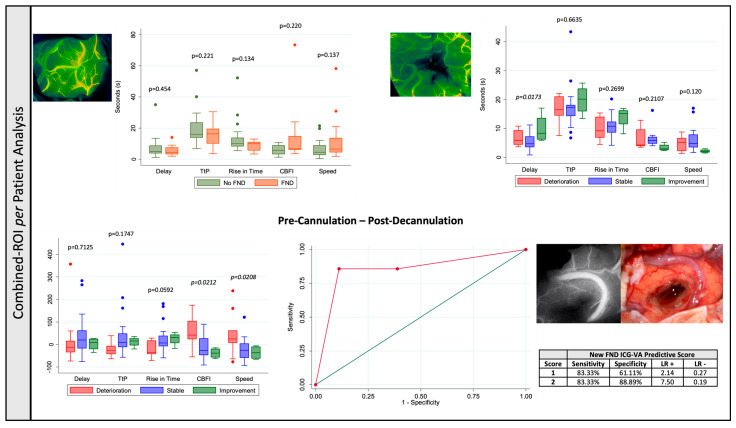
Combined Region-of-Interest per Patient Analysis—(**Top left**): Pre brain cannulation quantitative flow metrics and presence of preoperative focal neurological deficit. (**Top right**): Post brain decannulation quantitative flow metrics and the postoperative overall neurology. (**Bottom left**): Pre brain cannulation—post brain decannulation ICG flow metric percentage changes and overall neurology. (**Bottom right**): Receiver Operating Characteristic Curve Analysis using the flow metrics that showed statistical significance during the uni and multivariate analysis.

**Table 1 cancers-17-01392-t001:** Clinical characteristics of the studied population. * The WHO grading excluded patients with the diagnosis of metastatic disease. ** For the calculation of residual tumour volume and extent of resection, we excluded patients where the surgical procedure was a biopsy.

Demographics, Clinical and Neuropathology	
Age (y)	54.94 ± 15.32
Sex Male Female	20 (57.14%) 15 (42.86%)
Location Frontal Temporal Intraventricular Cerebellum Cingulate Parietal	12 (34.29%) 8 (22.86%) 6 (17.14%) 5 (14.29%) 3 (8.57%) 1 (2.86%)
Laterality Right Left	14 (40.00%) 21 (60.00%)
Neurological Examination Preoperative Neurological Deficit Postoperative Neurological Deficit Deterioration Stable Improvement Permanent Neurological Deficit (Preop + Post-op New Deficit)	13 (37.14%) 17 (48.57%) 10 (28.57%) 21 (60.00%) 4 (11.43%) 9 (25.71%)
WHO Grade * 1 2 3 4	6 (17.14%) 2 (5.71%) - 18 (51.43%)
Histology Glioblastoma Metastasis Meningioma Extraventricular Anaplastic Ependymomma Subependymmal Giant Astrocytoma WHO Grade 2 Astrocytoma WHO Grade 2 Central Neurocytoma	18 (51.43%) 9 (25.71%) 2 (5.71%) 1 (2.86%) 1 (2.86%) 2 (5.71%) 2 (5.71%)
Surgery	
Number of Regions-of-Interest Mean per patient	144 4.11 ± 1.57
Time of Brain Cannulation (min)	147.09 ± 71.91
Access Superior Frontal Sulcus Intraparietal Sulcus Superior Temporal Sulcus Longitudinal Sulcus	17 (48.57%) 7 (20.00%) 5 (14.29%) 6 (17.14%)
Length Tubular Retractor (mm) 50 60 75	2 (5.71%) 25 (71.43%) 8 (22.86%)
Imaging	
Depth of Lesion (mm)	3.72 ± 2.30
Preoperative Volume (cc)	29.92 ± 24.40
Residual Tumour Volume (cc) **	4.00 ± 9.94
Extent of Resection (%) ** GTR Near Total Resection Subtotal Partial Biopsy	88.27 ± 18.71 14 (40%) 4 (11.43%) 9 (25.71%) 5 (14.29%) 3 (8.57%)
Total Restriction to Diffusion (cc)	12.46 ± 20.72
Along-the-Path Restriction to Diffusion (cc)	5.16 ± 6.76

**Table 2 cancers-17-01392-t002:** Statistical analysis of region-of-interest based approach. Statistical analysis of the impact of preoperative focal neurological deficit (FND) in the pre-brain cannulation quantitative flow metrics and their variations in overall neurology.

Pre-Cannulation in ICG-VA Properties (s)	Focal Neurological Deficit (FND)			
	No FND	FND		*p* value
Delay	7.04 ± 5.95	5.07 ± 3.58		0.0147
Speed	70.15 ± 58.82	137.49 ± 156.53		0.0044
Time to Peak	20.27 ± 9.98	15.08 ± 7.33		0.0005
Rise in Time	13.35 ± 8.41	8.07 ± 3.59		<0.0001
Cerebral Blood Flow Index	6.14 ± 3.18	15.07 ± 20.81		0.0085
Post-Decannulation in ICG-VA properties (s)	Overall Neurology			
	Deterioration	Stable	Improvement	*p* value
Delay	8.73 ± 7.35	4.72 ± 3.51	9.36 ± 3.23	0.0001
Speed	49.98 ± 32.04	123.93 ± 128.86	37.93 ± 11.30	0.0001
Time to Peak	21.61 ± 11.00	17.05 ± 8.99	17.53 ± 3.27	0.1315
Rise in Time	14.69 ± 10.31	10.05 ± 5.92	10.8 ± 1.71	0.0016
Cerebral Blood Flow Index	5.53 ± 3.16	11.94 ± 16.92	5.78 ± 1.37	0.0057
Post–Pre-Cannulation Difference in ICG-VA properties (%)	Overall Neurology			
	Deterioration	Stable	Improvement	*p* value
Delay	140.77 ± 816.54	152.97 ± 458.70	−2.93 ± 34.84	0.6351
Speed	43.12 ± 80.60	−14.51 ± 57.80	−36.93 ± 31.33	<0.0001
Time to Peak	−20.16 ± 27.32	34.73 ± 143.83	14.20 ± 32.31	0.0406
Rise in Time	−21.97 ± 32.35	57.33 ± 219.90	26.53 ± 39.13	0.0552
Cerebral Blood Flow Index	50.40 ± 88.17	−2.70 ± 67.54	−38.98 ± 26.17	0.0005

**Table 3 cancers-17-01392-t003:** Statistical analysis of combined regions-of-interest *per* patient approach. Statistical analysis of the impact of preoperative focal neurological deficit (FND) in the pre-brain cannulation quantitative flow metrics and their variations in overall neurology.

Pre- Cannulation in ICG-VA Properties (s)	Focal Neurological Deficit (FND)			
	No FND	FND		*p* value
Delay	7.26 ± 7.16	5.57 ± 3.47		0.3602
Speed	65.47 ± 54.40	131.83 ± 158.19		0.1658
Time to Peak	19.52 ± 11.43	15.00 ± 7.40		0.1649
Rise in Time	13.43 ± 10.33	8.70 ± 3.09		0.0565
Cerebral Blood Flow Index	6.09 ± 2.91	15.25 ± 21.33		0.2089
Post-Decannulation in ICG-VA properties (s)	Overall Neurology			
	Deterioration	Stable	Improvement	*p* value
Delay	9.70 ± 9.59	4.58 ± 2.93	9.49 ± 3.15	0.0173
Speed	52.2 ± 39.22	118.05 ± 130.88	38.23 ± 12.75	0.1200
Time to Peak	23.01 ± 13.52	15.40 ± 8.68	17.74 ± 2.74	0.6625
Rise in Time	16.35 ± 13.72	9.61 ± 5.20	11.04 ± 1.40	0.2669
Cerebral Blood Flow Index	5.09 ± 3.26	13.15 ± 18.14	5.73 ± 1.39	0.2107
Post–Pre-Cannulation Difference in ICG-VA properties (%)	Overall Neurology			
	Deterioration	Stable	Improvement	*p* value
Delay	20.95 ± 123.94	42.54 ± 96.08	2.20 ± 28.21	0.7125
Speed	45.33 ± 94.28	−21.31 ± 46.96	−36.09 ± 30.28	0.0208
Time to Peak	−18.98 ± 29.95	45.78 ± 110.03	11.81 ± 23.27	0.1747
Rise in Time	−21.61 ± 33.65	27.71 ± 61.30	24.88 ± 30.27	0.0592
Cerebral Blood Flow Index	55.17 ± 77.45	−14.13 ± 47.43	−38.39 ± 22.19	0.0212

## Data Availability

Data about the technology used for trMIPS can be found on the following website: https://niconeuro.com/our-integrated-system/ (accessed on 1 March 2025).

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
