# Peer review of "Microvascular Cortical Dynamics in Minimal Invasive Deep-Seated Brain Tumour Surgery"

_cancers, 2025, doi:10.3390/cancers17091392_

Round 1
Reviewer 1 Report
Comments and Suggestions for Authors
The authors present an interesting work about the impact of perfusion reduction in deep tumor surgery. The idea is good, the presentation and methodology should be improved however.
I would like to point some issues I have:
1) some typos or missing letters should be corrected
2) the conclusion in the abstract is not a real conclusion. It is an argument for the validity of the results but not the conclusion of your work. So please change it.
3) How did you compare the ROIs? Which software?
4) How did you compare the ICG results? Is there a software for this?
5) Did you compare the surface cortex perfusion? What about the subcortical space?
6) A perfusion imaging would be nice, at least after surgery, to demonstrate the validity of your results.
7) Were the ROIs the same to all patients? Which were they and how did you decide about them? What was the contribution of the ROI analysis to the study? Please explain your primary idea for the study better in the text.
8) Please provide the images in better resolution.
9) Was the perfusion analysis on the tumor bed?
10) Prinz et al reported that FLOW 800 may detect procedure-related hemodynamic changes within the microcirculation and macrocirculation but should not be used as a stand-alone tool for quantitative flow assessment. (https://pubmed.ncbi.nlm.nih.gov/24335820/). Further scientific works report that FLOW 800 is helpful for the visualization of vessels during surgery but not appropriate for perfusion analysis. Please comment in your text. How do you support your results?
Author Response
Thank you very much for this opportunity to review our manuscript entitled Microvascular Cortical Dynamics in Transsulcal Tubular Retractor Assisted Minimal Invasive Parafascicular Surgery for Deep Seated Brain Tumours.
We have very honoured with the feedback provided by your esteemed panel of reviewers and your consideration to the work presented in the manuscript. We took into consideration carefully the suggestions and comments performed, and we extensively revised the manuscript to accommodate the feedback provided. Also, we reviewed the typos and the English format and layout to further improve its readability.
We believe the paper is its revised form has improved its quality, and our team hopes you find it suitable for publication in Cancers.
Yours sincerely,
The Authors
Comment 1: The authors present an interesting work about the impact of perfusion reduction in deep tumor surgery. The idea is good, the presentation and methodology should be improved however.
Authors: Thank you for considering the novelty of this work and we worked to improve the presentation of the paper and the way we explain the methodology used in this work.
I would like to point some issues I have:
Comment 1: some typos or missing letters should be corrected.
Authors: Thank you for highlighting this issue. We have extensively reviewed the manuscript, particularly the native English speakers in the authors list, to minimize and virtually correct all the grammatical and language-related errors that were present in the initial version.
Comment 2: the conclusion in the abstract is not a real conclusion. It is an argument for the validity of the results but not the conclusion of your work. So please change it.
Authors: Thank you for this comment. We agree that particularly the second sentence in the conclusion is interpretative and therefore needs to be amended. Therefore, the new conclusion in the abstract conclusion section is as follows:
“Microvascular flow dynamics impact trMIPS outcomes as an increase in the speed and CBFI after decannulation was related with worse neurological outcome.”
Comment 3: How did you compare the ROIs? Which software?
Comment 4: How did you compare the ICG results? Is there a software for this?
Authors: Thank you for these questions. We have decided to answer both these questions together as there are quite intermingled in the way we approached them in the methodology as well as the changes produced in the manuscript after providing this reply. The software used to delineate and compare the ROIs as well as to perform the ICG analysis and comparison of the results is embedded in the ZEISS KINEVO 900© microscope (Indocyanine green videoangiography FLOW 800 analysis). The software allows for retaining of the ROIs between different analysis in different time points and this was the main tool to ascertain the ROIs were the same. As brain shift and deformation (particularly after brain decannulation and tumour resection) are well known problems in brain tumour surgery, visual inspection of the anatomical documentation provided by intraoperative pictures was performed to address those sources of potential inaccuracies. Finally, these results were analysed with the statistical software mentioned in the methodology section - Stata 13.1 software© package (StataCorp LLC, Lakeway Drive College Station, Texas, USA).
To provide further clarification with regards to this methodology, we have done the following additions to the methods section:
“To guarantee that the same vascular areas were analysed, we maintained the same ROIs used in the pre-cannulation assessment for the post-decannulation assessment and performed visual inspection of the intraoperative images to ascertain they were in the same position to account for brainshift and deformation phenomena that occur in tumour surgery after resection.”
Comment 5: Did you compare the surface cortex perfusion? What about the subcortical space?
Comment 9: Was the perfusion analysis on the tumor bed?
Authors: Thank you for these questions. We have decided to answer them together given the overlap of the answer and the changes performed in the manuscript. All the analysis performed addressed the cortical structures. This is a consequence of the technique used to study this topic - ICG video angiography – which is well-validated for to address changes in the vascular dynamics that compromise filling (aneurysms) and vascularization of cortical areas. As far as it is our knowledge, this software - ZEISS KINEVO 900© microscope Indocyanine green videoangiography FLOW 800 analysis – has not been validated, regularly used and approved for subcortical / white matter vascularization assessment. Therefore, we focused on the translation and interpretation of cortical data which has been previously validated in the neurovascular field instead of starting by exploring novel applications. Also, given the limitations provided by depth of focus, it is challenging if not impossible for deep seated lesions, to assess in the same data acquisition cortical and subcortical/tumour bed ROI. This would then imply multiple ICG administrations and the potential effect of ICG contamination from one administration to the others. Finally, the subcortical/tumour bed flow assessment would necessarily imply the persistent of the tubular retractor in situ. This would raise concerns about the significance potential findings as these values would necessarily significantly change after tubular retractor removal and it would be difficult to extrapolate the subcortical flow values for these metrics in the absence of the tubular retractor.
To improve the clarification of this point in the manuscript, we added the following sentences in the limitation section:
“Only cortical microvascularization was analysed and not data reported to the subcortical compartment / tumour bed is provided in this study. The reasons behind this decision were: a) the lack of previously data reporting/validating subcortical perfusion assessment using FLOW 800 ICG-VA analysis; b) the technical challenge of providing depth focus to acquire ROIs in cortical and subcortical areas in the same ICG analysis (which would imply multiple administrations and the potential effect of cross ICG contamination in between them); and c) the artificial effect of the tubular retractor presence when acquiring subcortical data which would challenge the significance of the results obtained.”
Comment 6: A perfusion imaging would be nice, at least after surgery, to demonstrate the validity of your results.
Authors: This is a very good point. This work was performed and approved as an audit whereby no changes were introduced in the clinical management of these patients. Unfortunately, perfusion imaging in the postoperative setting after brain tumour surgery is not part of the clinical protocol approved in our institution and not standard of care in brain tumour assessment immediately after surgery as per NICE guidelines. This are the reasons why this imaging methodology was not acquired and analysed. However, we agree this should be the next step in this research field as changes in perfusion sequences could be a better correlate with the intraoperative tools rather than the diffusion sequences we analysed. We addressed this issue in the limitation section as follows:
“Finally, no immediate postoperative perfusion imaging was acquired as these sequences are not included in the protocol for brain tumour evaluation in this time point. Future studies should consider these sequences and their correlation with the studied ICG-VA-derived metric and ascertain if they provide a better surrogate of intraoperative findings when compared with diffusion imaging.”
Comment 7: Were the ROIs the same to all patients? Which were they and how did you decide about them? What was the contribution of the ROI analysis to the study? Please explain your primary idea for the study better in the text.
Authors: Thank you for this comment. The ROIs were not the same for all patients. Their variability is explained by the different entry points used for the minimal invasive parafascicular approach (different lobar / sulci regions). Also, the number of ROIs per patient was different and this is the reason behind us present the average number of ROIs per patient with a range. Two main reasons explain the variability in number of ROIs: the retrospective nature of the study in an audit setting (prospective studies would allow for a fixed number of ROIs to be determined in the inclusion criteria) and the craniotomy size which would vary according to the other surgical adjuncts planned to be used during surgery (intraoperative ultrasound and intraoperative neuromonitoring and mapping tools mainly). The craniotomy size determined the amount of perisulci arteries available to monitoring with the appropriate ROIs placed at the beginning and end of surgery. This ROI-per-patient variability led our group to perform a the Combined-ROI analysis. Our aim with this second approach was to minimize the impact of outliers particularly in patients with a reduced amount of ROIs analysed and provide a more global assessment of each metric per patient. This will bring us to the final point you raise in this question about the aim of the study among all these analysis and metrics. The aim of this project was to assess the utility of this well-established ICG-VA FLOW 800 analysis metrics in tubular retractor assisted surgery for deep seated brain tumours. Single-ROI analysis is simple and quick and provide immediate feedback to the surgeon with no need for offline mathematic assessment whereas a Combined-Roi per patient analysis require off line averaging. This makes the single-ROI analysis appealing for surgical use. However, we had to confirm if the results provided with this approach were robust and reproducible enough when averaged among all the available ROI per patient. This is the added value of the combine-ROI per patient analysis.
To improve the explanation of our reasoning, we added the following sentences to the Methods and the Limitations and Strengths sections:
Methods:
“The location of the selected ROIs were determined by the chosen surgical corridor (sulcus of interest and perisulci arteries available to monitor). The number of ROIs selected was also different and determined by the size of the craniotomy which showed variability according to the other surgical adjuncts planned to be used during surgery (intraoperative ultrasound and intraoperative neuromonitoring and mapping tools, mainly).”
“This combined-ROI per patient-based analysis aim to provide robustness to the region-of-interest-based approach allowing to minimize the effect of outliers in the later approach.”
Limitations and Strengths:
“This is a retrospective study with the inherent limitations to this study type design, particularly incomplete data that lead to patient exclusions from the analysis and variability of number of ROIs selected per patient (reason why we present the average and range of ROIs per patient).”
“ROI-based findings (validated by the combined-ROI per-patient approach) can provide an easy and assessable tool to the surgical team to integrate in multimodal risk stratification methods of neurological outcome after surgery (as for example, intraoperative neuromonitoring and preoperative mapping data).”
Comment 8: Please provide the images in better resolution.
Authors: Thank you for highlighting this issue. We have addressed the image resolution in the revised uploaded version of the paper. However, the processed images with the ICG data using the FLOW 800 algorithm have less anatomical detail when compared with the optical images which we are not able to correct as they are the output of the Zeiss algorithm.
Comment 10: Prinz et al reported that FLOW 800 may detect procedure-related hemodynamic changes within the microcirculation and macrocirculation but should not be used as a stand-alone tool for quantitative flow assessment. (https://pubmed.ncbi.nlm.nih.gov/24335820/). Further scientific works report that FLOW 800 is helpful for the visualization of vessels during surgery but not appropriate for perfusion analysis. Please comment in your text. How do you support your results?
Authors: Thank you for this remark. We do agree that the vast majority of the literature published on this topic is focused mainly on the enhanced anatomic properties of this technique and not in its quantitative perfusion analysis capacity. Also, the neurovascular field has been leading and driving the research in intraoperative quantitative perfusion analysis. Prinz and his group compared the FLOW 800 quantitative analysis with intraoperative eco-doppler and cortical laser speckle imaging. They concluded that FLOW 800 quantitative analysis changed with bypass surgery in the expected way with improvement of blood flow with a successful bypass but this change did not show direct correlation with the other 2 techniques. In this study, we did not compare the FLOW 800 changes with other techniques. This is a limitation of this study and we believe it is a very valid point as a future pathway of research. We have included this topic in our limitation section. On the other hand, we believe that Prinz study support our line of research and findings as FLOW800 technique was also able in their study to detect significant (and expected) changes with their surgical technique under scrutiny – bypass. It is clear that we still need to learn more about the potential correlation between these intraoperative fluorescence-based and some classic methods such as the ecodoppler. However, as Prinz showed in the neurovascular pathology treated with bypass, we show in this paper that this technique can provide valuable perfusion information during tubular retractor assisted brain tumour resection. We have changed the manuscript as follows to address these topics:
Discussion:
“From a technical perspective, this study comes alongside with others from the neurovascular field that support that ICG-VA FLOW 800 assessments can identify and provide quantitative metrics for both micro and macrovascular changes after successful intraoperative treatment.”
Limitations and Strengths:
“Also, it is important to further understand the correlation of these intraoperative adjunct with other tools available for micro and macrovasculatization assessment, such as the ecodoppler as previous studies failed to provide a reliable correlation.”
Thank you again for your work and your comments.
The authors
Reviewer 2 Report
Comments and Suggestions for Authors
Comments to authors-2025-03-10
Submission ID; cancers-3534574
Thank you very much to give me an opportunity to review this manuscript.
Authors described that correlation between alteration in microcirculation and the frequency of neurological deficits following tubular retractor assisted surgery for deep seated brain tumors including glioblastoma and brain metastases.
They demonstrated an increase in speed and cerebral blood flow index related with deterioration of neurological presentation and postoperative neurological functional outcome in both pre- and post-cannulation.
Given the difficulty in interpretation of data presented in the current article, I have several questions.
#1. What, do authors think, is the advantage for using tube retractor for surgery for deep seated brain tumors? Will parameters for validation in the current study including speed, time to peak, rise time, and cerebral blood flow index, be different with or without tube retractor?
#2. Did authors think that alteration of these parameters including speed, time to peak, rise time, and cerebral blood flow index, in surgery with tube retractor compared with surgery without using tube retractor?
#3. To my opinion, size of the tumor and duration of brain cannulation could be confounding factors. I am curious whether neurological outcome would be different between long and short duration of surgery. I wish authors demonstrate this type of data of comparison between short and long duration of surgery stratified based on cut-off value as average.
#4. I was wondering why increase cerebral blood flow after retractor removal reveal worse neurological outcome. Is deterioration of neurological function related with ischemia? I would like to ask authors to elucidate the mechanism for rescuing neurological outcome when tube retractor was used.
Comments on the Quality of English Language
"Assessment was performed" should be changed to ".... was assessed".
Author Response
Thank you very much for this opportunity to review our manuscript entitled Microvascular Cortical Dynamics in Transsulcal Tubular Retractor Assisted Minimal Invasive Parafascicular Surgery for Deep Seated Brain Tumours.
We have very honoured with the feedback provided by your esteemed panel of reviewers and your consideration to the work presented in the manuscript. We took into consideration carefully the suggestions and comments performed, and we extensively revised the manuscript to accommodate the feedback provided. Also, we reviewed the typos and the English format and layout to further improve its readability.
We believe the paper is its revised form has improved its quality, and our team hopes you find it suitable for publication in Cancers.
Yours sincerely,
The Authors
Reviewer: Thank you very much to give me an opportunity to review this manuscript.
Authors described that correlation between alteration in microcirculation and the frequency of neurological deficits following tubular retractor assisted surgery for deep seated brain tumors including glioblastoma and brain metastases.
They demonstrated an increase in speed and cerebral blood flow index related with deterioration of neurological presentation and postoperative neurological functional outcome in both pre- and post-cannulation.
Authors: Thank you very much for the very accurate summary of the key findings of our work.
Reviewer: Given the difficulty in interpretation of data presented in the current article, I have several questions.
Comment #1. What, do authors think, is the advantage for using tube retractor for surgery for deep seated brain tumors? Will parameters for validation in the current study including speed, time to peak, rise time, and cerebral blood flow index, be different with or without tube retractor?
Response #1: Thank you very much for this question. We have not explored extensively the benefits of the tubular retractor surgery in this manuscript as the main topic of the paper was to address the findings regarding the perfusion analysis related the tubular retractor technique and not the indications. However, we do agree with you that potential readers that might not be familiar with the technique significantly benefit from further explanation about the benefits and rational behind this technique. In our first version of the paper, this explanation was summarized in the following sentence:
“In this context, transsulcal tubular retractor-assisted minimal invasive parafascicular approach (ts-trMIPS), has gained significant popularity as it is a tissue-sparing technique preserving cortical and subcortical structures from the brain surface to the lesion”
We have fully considered your question and we expanded this section with the following sentences in the Introduction:
“As a transsulcal approach, it potentiates the preservation of the main projection and association fibers that originate from the crest of the gyrus. This benefit is combined with stable retraction and pressure over the surrounding tissue provided by the tubular retractor. Previous publications have shown a decrease in the chances of perisurgical corridor contusion and laceration when compared with blades or brain spatulas.”
Regarding your second question, at the moment we have no data to confirm or deny and that is a current line of research. We believe that data coming from non-tubular retractor surgery can help us to understand if these changes are related with the technique or tumour location. We have explored this topic in the Discussion section as follows:
“Particularly, it is important to understand how these metrics behave in superficial non-tubular retractor assisted surgery. This will enhance our comprehension about the main drivers for these reported changes: tubular retractor surgical adjunct versus tumour location.”
Comment #2. Did authors think that alteration of these parameters including speed, time to peak, rise time, and cerebral blood flow index, in surgery with tube retractor compared with surgery without using tube retractor?
Response #2: Thank you very much for this question. We tried to address this question in an holistic manner included in the second point from your previous question and added the following statement to the Discussion section in the paper:
“Particularly, it is important to understand how these metrics behave in superficial non-tubular retractor assisted surgery. This will enhance our comprehension about the main drivers for these reported changes: tubular retractor surgical adjunct versus tumour location.”
We believe that currently we have no data about non-tubular retractor surgery and therefore any further comments would be speculative given the absent data in neuro-oncology in the literature to the best of our knowledge. We do hope you understand this limitation at the moment.
Comment #3. To my opinion, size of the tumor and duration of brain cannulation could be confounding factors. I am curious whether neurological outcome would be different between long and short duration of surgery. I wish authors demonstrate this type of data of comparison between short and long duration of surgery stratified based on cut-off value as average.
Response #3: Thank you for your comment and suggestions. We have performed and presented the adjusted analysis for initial tumour volume (pre-cannulation metrics) and tumour residual (post-cannulation and pre cannulation - post decannulation) for all the presented metrics and these results are presented in the Results section. We agree with your suggestion of further interrogating the data about a potential cut-off that would change the impact of each metric and neurological outcome. We have calculated the median for both the single ROI distribution and in the combined ROI per patient analysis. For the single ROI approach, we studied the impact of the time of cannulation in both the post-decannulation and pre-cannulation post-decannulation differences. For the combined ROI per patient analysis, we have performed the same analysis and further assessed the impact of this division on the neurological outcome and radiological diffusion restriction to diffusion in the around the surgical corridor. Our findings suggest no impact of time of cannulation in both neurological and radiological outcomes and the indication provided in the single-ROI analysis were not confirmed at patient level. Therefore, we do not think time of cannulation is a significant driver of the results encountered in the ICG-VA metrics but this would require further validation in a larger cohort.
We summarized these in the Methods and Result section and discussed them in the Discussion section as follows. Also, we provided a Supplementary Material Table to further illustrate these findings.
“Finally, we performed a subgroup analysis according to the median time of cannulation to assess the impact produced by the period of the time the cortical-subcortical structures are under the physical stress of the tubular retractor on the neurological and radiological outcomes.”
“A subgroup analysis according to the time of brain cannulation (cut-off: median of distribution) showed no statistically significant impact of time of brain cannulation in both new focal neurological deficit after surgery (p=0.915) and restriction to diffusion around the surgical corridor (p=0.176). With regards to the ICG-VA-derived metrics, the single ROI analysis showed a decrease of speed (p=0.044) and TtP (p=0.039) post decannulation and an increase in the delay (p=0.038) comparing pre-cannulation and post-decannulation in the group of patients with longer cannulation times. However, the combined-ROI per patient analysis did not confirm these findings (p>0.05). (Supplementary Material 1)”
“Even though single-ROI analysis showed some trends between time of cannulation and ICG-VA derived metrics, none of these were confirmed at patient level and therefore they are potentially not relevant in driving the observed micro and macrovascular changes.”
Comment #4. I was wondering why increase cerebral blood flow after retractor removal reveal worse neurological outcome. Is deterioration of neurological function related with ischemia? I would like to ask authors to elucidate the mechanism for rescuing neurological outcome when tube retractor was used.
Response #4: This is a very good point and completely open for discussion. We discussed in the Discussion section 3 potential mechanisms why we believe this phenomena occur:
- Compensatory Vasodilation distal to ischemic-related vasoconstriction: this could partially explain the increase in the cerebral blood flow.
“These findings were observed in animal studies in penumbra areas where the capillaries that remained perfused after an ischemic insult developed vasoconstriction and increased flow as a consequence of compensatory arterial vasodilation [38,39].”
- Neurovascular uncoupling and spreading depolarizations: this can explain neurological deficit in the absence of MRI-documented ischemia.
“An impaired neurovascular coupling in the peri-ischemic area may play a crucial role in the poor outcome even in the absence of ischemia [38,42,43]. Driving changes in cortical microvascular flow dynamics could also be occult spreading depolarisations (SD) – well known to incite prolonged changes in the local microcirculation [44,45].”
- Absence of significant differences in the subcortical flow as a consequence of a single vessel occlusion in the microvasculatization network: this can explain neurological deficit in the absence of MRI-documented ischemia.
“Even though the subsurface flow is not significantly affected by a single vessel occlusion within the network, this will cause cortical subcortical ischemia which supports our findings of dissociation between blood flow assessment and ischemia in the postoperative MRI scan.”
Regarding the second aspect of your question, this was an observational study and therefore our ethics under the neurosurgery governance team did not contemplate intervention and/or manipulation of treatment targets in the postoperative care. Having said that, we hypothetize if treatments to maintain euvolemia and higher blood pressure targets could benefit patients with confirmed post-operative neurological deficits and post-decannulation confirmation of high speed and higher cerebral blood flow. However, these interventions need to be balanced with the risk of postoperative heamatoma and/or haemorrhagic transformation within the perisurgical corridor. In order to address this comment and future directions in treatment-driven research, we have added the following statement to the Limitations and Strengths section:
“These data can also improve patient selection for future studies aiming to intra and/or postoperative treatments targeting the impact cerebral blood flow and neurovascular coupling - euvolemia or flow augmentation via manipulation of postoperative targets for blood pressure management - in neurological outcomes.”
Thank you again for your work and your comments.
Round 2
Reviewer 1 Report
Comments and Suggestions for Authors
Thank you for the changes and the explanation to my remarks.
I would change the order of the words in your title, you can consider if you want to adopt it:
Microvascular Cortical Dynamics in Minimal Invasive Tubular Retractor Assisted Transsulcal Parafascicular Surgery for Deep Seated Brain Tumours
or just
Microvascular Cortical Dynamics in Minimal Invasive Deep Seated Brain Tumour Surgery
Otherwise I am satisfied, good work to you all.
Author Response
Thank you very much for the comments and the suggestions, also regarding the title which we have changed accordingly:
the new title is now: Microvascular Cortical Dynamics in Minimal Invasive Deep Seated Brain Tumour Surgery
thank you again for your support
the Authors.
Reviewer 2 Report
Comments and Suggestions for Authors
Thank you very much to give me an opportunity to re-review this manuscript.
I appreciated authors to re-write revised manuscript and to respond my comments sincerely point by point.

Author Response
Dear reviewer,
we would like to sincerely thank you very much for your comments, your revision, your help and your support.
best regards
the Authors